# Association of Parathyroid and Differentiated Thyroid Carcinomas: A Narrative Up-To-Date Review of the Literature

**DOI:** 10.3390/medicina58091184

**Published:** 2022-08-30

**Authors:** Razvan Simescu, Miana Pop, Andra Piciu, Valentin Muntean, Doina Piciu

**Affiliations:** 1Humanitas Hospital Cluj-Napoca, 400664 Cluj-Napoca, Romania; 2Department of Surgery, University of Medicine and Pharmacy Iuliu Hatieganu Cluj-Napoca, 400347 Cluj-Napoca, Romania; 3Department of Medical Oncology, University of Medicine and Pharmacy Iuliu Hatieganu Cluj–Napoca, 400347 Cluj-Napoca, Romania; 4Institute of Oncology Prof. Dr. I. Chiricuta Cluj-Napoca, 400015 Cluj-Napoca, Romania; 5Doctoral School, University of Medicine and Pharmacy Iuliu Hatieganu Cluj-Napoca, 400347 Cluj-Napoca, Romania

**Keywords:** parathyroid carcinoma, well differentiated thyroid cancers, tumor association, preoperatory suspicion, en bloc resection

## Abstract

**Aim:** Parathyroid carcinoma (PC) is a rare endocrine malignancy that represents 0.005% of all malignant tumors. Associated PC and differentiated thyroid carcinoma (DTC) is an exceptionally rare condition, and the preoperative diagnostics and proper treatment are challenging. Almost all PCs and the majority of DTCs are diagnosed postoperatively, making correct surgical treatment questionable. Specific guidelines for parathyroid and thyroid carcinomas association treatment are lacking. The purposes of our study were to identify the association between parathyroid and thyroid carcinomas, to analyze the available published data, and to evaluate the possible relationship between preoperative diagnostic and surgical decision-making, and outcome-related issues. **Material and methods:** We performed a literature review of several databases from the earliest records to March 2022, using controlled vocabulary and keywords to search for records on the topic of PC and WDTC pathological association. The reference lists from the initially identified articles were analyzed to obtain more references. **Results:** We identified 25 cases of PC and DTC association, 14 more than the latest review from 2021. The mean age of patients was 55, with a female to male ratio of about 3:1. Exposure to external radiation was identified in only one patient, although it is considered a risk factor the development of both PC and DTC. The preoperative suspicion of PC was stated by the authors in only 25% of cases, but suspicion based on clinical, laboratory, ultrasound (US), and fine needle aspiration (FNA) criteria could have been justified in more than 50% of them. With neck ultrasound, 40% of patients presented suspicious features both for PC and thyroid carcinoma. Intra-operatory descriptions of the lesions revealed the highest suspicion (83.3%) of PC, but en bloc resection was recommended and probably performed in only about 50% of the cases. Histopathological examinations of the thyroid revealed different forms of papillary thyroid carcinoma (PTC) in most cases. Postoperative normocalcemia was achieved in 72% of patients, but follow-up data was missing in about 25% of cases. **Conclusion:** Associated PC and DTC is an exceptionally rare condition, and the preoperative diagnostic and treatment of the patients is a challenge. However, in most cases pre- and intraoperative suspicious features are present for identification by a highly specialized multidisciplinary endocrine team, who can thus perform the optimal treatment to achieve curability.

## 1. Background/Introduction

Parathyroid carcinoma is a cause of primary hyperparathyroidism (PHPT), PA (parathyroid adenoma), and PH (parathyroid hyperplasia), but less than 1% of PHPT cases are due to PC [1,2]. The peak incidence of PC occurs in the fifth decade of life, and there is no gender predilection [3]. DTCs arising from thyroid follicular epithelial cells are by far the most frequent thyroid cancers. PTC accounts for approximately 80–85% of DTC cases, while FTC accounts for approximately 12% [4]. In recent years, an increase in the incidence of thyroid carcinoma, in particular mPTC [5] has been observed, more marked among middle-aged women (aged 35–64 years) [6]. The incidence rates of DTC increased sharply in women during reproductive age, then declined and equalized with men by 85 years of age [7]. The reported female to male ratio was 2.9:1 [5,8].

Association of parathyroid and thyroid carcinomas is extremely rare [9,10,11]. When it occurs, PC is more likely to associate with DTC, mainly PTC [12]. Most PCs are sporadic, but they can also occur in the context of genetic syndromes. While some authors considered the combined occurrence of parathyroid and thyroid cancers a coincidence [13], others identified hypercalcemia or factors like epithelial and insulin growth factor as having a goitrogenic effect [14,15,16]. Sporadic PC and DTC have both been linked with exposure to external radiation [17,18].

In terms of treatment recommendations, specific guidelines for synchronous parathyroid and thyroid carcinomas treatment are absent. To date, the only accepted curative treatment for PC is en bloc resection of the affected parathyroid gland, with hemithyroidectomy of the ipsilateral lobe and surrounding adherent tissue including enlarged lymph nodes preoperatively diagnosed or identified during surgery [19,20,21]. DTC treatment is also surgical, and the extent of resection is determined by the extent of local and regional spread of the disease [22].

Almost all PCs and the majority of DTCs have been diagnosed postoperatively, making proper surgical treatment questionable. Preoperative suspicion of PC is difficult and rare, partly because clinical manifestations are often due to other causes of hyperparathyroidism [11,23,24]. DTC is often clinically silent, and half of all cases were found to present as incidental findings during physical examination or on neck US, or as previously unsuspected histological findings [25].

No specific imaging method can differentiate PC from benign parathyroid disease [21], but neck US has the greatest potential to raise both DTC and PC suspicion [26]. While FNA is useful in DTC preoperative diagnosis, it should be avoided in parathyroid lesions [27,28].

Intra-operative findings have also been described that raise the suspicion of PC [29,30]. However, a low percentage of patients were reported to have undergone en bloc resection, most likely because their PCs were not identified either preoperatively or even intraoperatively [31,32].

The extremely rare and problematic association between parathyroid and thyroid carcinoma prompted us to perform an up-to-date literature review to identify a possible relationship between preoperative diagnostic workup data, surgical decision-making, and outcome related issues. Recent reviews by Lam-Chung et al. [9] in 2020 and De Falco et al. in 2021 [10] documented confusing numbers of case reports (15 and 11 cases, respectively) with only 10 of them being superimposable. Moreover, the cases and some of their data were presented in the form of tables providing only partial data, for comparison with the author’s case presentation.

Due to the rarity of parathyroid carcinoma and of the association with thyroid carcinoma, there are neither guidelines nor standard recommendations, and there have been no extensive randomized studies. Therefore, the aim of the study was to review the majority of publications in this field and provide an overview of the most common diagnostic and treatment strategies. This review can offer a comprehensive picture of the challenges faced during preoperative diagnosis, intraoperative decision-making, proper treatment, and case documentation, to enable improvement in future studies and patient treatments.

## 2. Methods

We performed a comprehensive search of databases including PubMed, Google Scholar, and Web of Science, from the earliest records in 1979 to August 2022. The search was carried out independently by two of the authors, using controlled vocabulary and keywords to search for records on the topic of PC and DTC pathological association. When the potentially eligible papers were retrieved, the full text publications were evaluated for eligibility.

Our focus was on information regarding demographics, relevant signs and symptoms, patients’ relevant medical and family history, preoperatory laboratory and imaging findings, indications for surgery, intraoperative findings and descriptions, surgical treatment details, final diagnostics, outcome, and follow-up. Exclusion criteria were a language other than English, inaccessibility of the full text, no case presentation, and not enough data presented. The workup of selecting eligible papers for our review is schematically presented in Figure 1.

Relevant data were independently retrieved by two of the authors, who are certified endocrine surgeons, and then confronted and analyzed with additional input from two more experts, i.e., an endocrinologist and a nuclear medicine physician.

## 3. Results

According to our literature review, 25 cases of parathyroid and thyroid carcinoma association were identified and analyzed, see Table 1. Mean age of all the patients was 54.88 years (range: 21–89), with means of female and male patients at 56.37 (range: 29–89) and 51 (range: 21–68), respectively, and 60% of the cases were over 50 years old. There was a predominance of female cases (19 cases, 76%), compared with male (six cases, 24%). 

Different symptoms of hyperparathyroidism were present in 62.5% of the 24 cases with clinical data available. Most patients had no irradiation history (92.86%), nor any relevant family history or genetic syndrome data (94.44%). In 9 cases (36%) the genetic syndromes were ruled out based on clinical, laboratory, and imagistic screening. Only one case had genetic testing for relevant family syndrome genes.

We defined the patients as having “prolonged hypercalcemia” when a long-lasting primary hyperparathyroidism (PHPT) history was stated, or when signs of possible long-lasting hypercalcemia (severe bone disease, progressive long-standing renal impairment, peptic ulcer disease, neurocognitive deficits) were described.

We defined as “data suspicious for PC” any signs and symptoms of severe hypercalcemia (nausea, vomiting, dehydration, polyuria, polydipsia, confusion, lethargy, constipation, peptic ulcer, etc.) and/or severe bone disease (severe osteoporosis, osteitis fibrosa cystica, pathologic fracture) and/or progressive renal impairment (chronic kidney disease, nephrocalcinosis) and/or neurocognitive deficits and/or palpable hard neck mass. Clinical data suspicious for PC was found in 16 patients (64%), while10 patients (36%) had no such clinical suspicion. Of those with no suspicion, seven patients were asymptomatic and three had only a neck mass palpable.

Suspicious elevated serum calcium levels (≥14 mg/dL) were found in nine patients (39.13%), with 14 patients (60.87%) having elevated calcium levels but <14 mg/dL, and one patient (4.35%) presenting normal values. Most of the patients (82.61%) presented highly elevated PTH levels (≥5–10-fold normal value), suspicious for PC, while four cases (17.39%) had elevated but non-suspicious PTH. One patient had no available data regarding calcium, and two regarding PTH. Non-functioning PCs with normal or slightly elevated calcium and PTH were mentioned in 4 cases.

In terms of bone disease in PC patients, parameters such as alkaline phosphatase, PO4, and renal function parameters are missing from many reports, so we decided not to include data referring to serum level creatinine or GFR.

Ultrasound (US) of the neck was not available in two cases (8%), and two patients had undergone previous thyroidectomies. Parathyroid or thyroid lesions were not identified by US in five case (21.47%) and one (4.76%) case, respectively. Parathyroid “suspicious lesions” for carcinoma were heterogeneous cystic structures with irregular or ill-defined borders and intra-nodular calcifications or signs of local invasion. Parathyroid lesions suspicious for PC were described in 50% (9 patients) of the US- identified parathyroids, the other half being diagnosed as possible Pas (18 patients in total). Associated or isolated thyroid lesions were found in 20 patients, with suspicious features being present in 15 (75%) of them, all without extrathyroidal or lymph node extension.

By US, concomitant parathyroid and thyroid lesions were identified in 15 (60%) patients, of whom six (40%) had suspicious features both for PC and thyroid carcinoma.

Fine needle aspiration (FNA) was performed on parathyroid glands for diagnostic reasons on a total of four lesions (16%). Thyroid-nodule FNA was not performed or not stated in 10 cases (40%). Of the other 15 cases which featured US suspicious thyroid nodules, FNAs were performed in only 10. Papillary thyroid carcinoma (PTC) was diagnosed by means of FNA in only three of these cases (30%), one of which was a double carcinoma. The other thyroid-nodule FNAs (seven cases, 70%) were either benign or non-conclusive.

In 21 cases (84%), ^99m^Tc scintigraphy with MIBI (Tc-99m methoxy–isobutyl–isonitrile) (planar or SPECT/CT) was carried out to localize the affected parathyroid gland, and 17 glands (80.95%) presented radiotracer uptake. In three of these cases (17.65%) the gland had not been identified by previous neck US. In four cases (19.05%) the examination was unable to localize the affected parathyroid gland. In one case, neither ^99m^Tc scintigraphy nor neck US were able to identify the pathological parathyroid.

“Preoperatory PC suspicion” refers to the cases in which the authors took into consideration and mentioned the PC diagnosis. A preoperative PC suspicion was mentioned in 24% (six out of 25) of the cases.

We defined “possible preoperatory PC suspicion” as at least two cumulative suspicions for PC based on available clinical-, laboratory-, US-, or FNA-based data. On these premises, in 56% of the cases (14 out of 25) the preoperatory suspicion for PC could have been justified.

“Intraoperatory PC suspicion” was based on the authors’ descriptions of suspicious features for PC. Data was available in 72% of the cases and it revealed the highest suspicion (83.3%) on PC. En bloc resection was performed and stated only in 52% of the cases (13 out of 25). In the remaining 12 cases (48%), en bloc resection was either not performed (including two cases with previous total thyroidectomies), or not mentioned.

There was a preponderance of left side parathyroid lesions (12 cases, 60%), of which more than two thirds were inferior glands. Of note is one case with an ectopic left mediastinal parathyroid gland that was diagnosed and treated in the recent past by concomitant PC-NMTC operation (synchronous double PC).

Mean maximal diameter of recorded PC was 3.15 cm (range: 1.2–5 cm). Definitive PC diagnosis was achieved through histopathological examination of the resected specimens, and immune staining was additionally performed in seven cases (28%). PC-positive surgical margins were present in two of the histopathological examinations. Micropapillary (mPTC) and papillary thyroid carcinoma (PTC) were present in 23 of the 25 patients, while follicular thyroid carcinoma (FTC) and Hurthle cell carcinoma (HCC) were identified in three patients and one, respectively. Multiple DTCs were found in 52% of patients (13 out of 25 cases). Of these, the majority were mPTC (five out of 11 cases), followed by mPTC associated with PTC (three cases), and mPTC associated with FTC and with HCC (each with one case).

Isolated single-type DTCs were identified in 48% of patients (12 out of 25 cases) with PTC and mPTC having equal shares (50% respectively).

Associated parathyroid disease was documented in five of the 25 cases, as two parathyroid adenomas (PA), two cases of parathyroid hyperplasia (PH), and one ectopic mediastinal PC, see Table 2.

Postoperative outcomes showed normal serum calcium level achieved in 72% of the patients (18 out of 25 cases). Persistent and recurrent disease were documented in a total of seven patients. Persistent disease was present in three of the patients (12%) and recurrence in four (16%).

Follow-up data was missing in about 25% of cases. The follow-up period was longer than two years for 11 of the total 25 patients (44%). One of the three patients with less than three months of follow-up died three weeks after surgery, due to uncontrolled hypercalcemic crisis.

There were no reported cases of postoperative hypocalcemia.

## 4. Discussion

The association of PC and DTC is an exceptionally rare clinical presentation, and no guidelines for diagnostic and treatment strategy are currently available.

Our review identified a total of 29 cases of associations between PC and DTC [9,10,11,12,21,33,34,35,36,37,38,39,40,41,42,43,44,45,46,47,48,49,50,51,52], but four of these we did not include in our analysis because they were not detailed as case presentations [53,54,55], or we could not obtain a full text article [56]. Compared with the last published review by de Falco et al. [10] from 2021, we were able to identify and analyze 14 more cases.

In our review, we identified that the mean age of patients was 54.88, but when we compared it by gender, we found that female mean age was higher than male mean age (56.37 vs. 51). This was probably because in most cases PC was the condition that determined the patients to seek medical attendance, and not the associated DTC. The predominance of female compared to male patients (approximately 3:1), more in accordance with DTC trends than with those of PC, is an intriguing feature of the pathological association.

### 4.1. Etiopathology of PC and DTC

The etiopathology of neither PC nor DTC is well-known. Most PCs are sporadic, but they can also occur in the settings of genetic syndromes, such as hyperparathyroidism-jaw tumor syndrome (HPT-JT), multiple endocrine neoplasia type 1 (MEN 1) and type 2A (MEN 2A), or familial isolated hyperparathyroidism [57,58,59]. Genetic syndromes could be suspected based on family history and on clinical, laboratory, or imagistic findings of associated diseases. Definitive exclusion of syndromes should be based on genetic testing of relevant genes such as CDC 73, CDKN1B, MEN 1, RET, etc. [34]. In 72% of the analyzed cases, family history was mentioned, but only the case presented by Kern et al. [48] had a possible relevant history, albeit without having been further investigated. Screening by non-genetic means was carried out in 36% of the cases, and only the case from Edafe et al. [34] was genetically tested by gene sequencing and dosage analysis. None of the cases were stated to be syndromic.

Previous studies found an increased risk of benign parathyroid disease and concurrent thyroid disease in patients who had been exposed to external radiation (especially of the head and neck) [60,61]. Sporadic PC and DTC have both been linked with exposure to external radiation [17,18], but only one case in our review was found to have had external radiotherapy, 19 years before the diagnosis of associated PC and DTC.

Reports have also suggested that long-standing secondary HPT or end-stage renal disease could be associated with increased risk of PC [17,24]. In our review, we did not identify any patients with secondary HPT or end-stage renal disease. Instead, in 55% of the cases we identified clues of possible long-standing hypercalcemia in the form of relevant renal (e.g., nephrolithiasis, nephrocalcinosis, reduced renal parenchymal index) and/or skeletal involvement (e.g., osteoporosis, osteitis fibrosa cystica, subperiosteal resorption, “salt and pepper” skull, or pathologic fractures). Also, three of these patients had over six-year long histories of hypercalcemia.

### 4.2. Clinical Manifestations

It is important to underline that these patients were not really asymptomatic: bone markers, the level of calcium, and osteoporosis were important signs that in many cases had been neglected for years. Meanwhile, patients with severe vitamin-D deficiency have very high levels of PTH in the presence of serum calcium levels in the upper range, and these levels were not been properly evaluated in order to rule out PHPT.

DTC is largely asymptomatic [25]. PC’s clinical manifestations are superimposable on those of other causes of PHPT and typically display an indolent course [62]. Still, PC should be suspected in all PHPT cases with fast onset or marked hypercalcemic symptoms or signs (such as anxiety, depression, weakness, weight loss, bone and renal disease, abdominal pain, nausea, pancreatitis, or peptic ulcer), hypercalcemic crisis, and/or a palpable neck mass [63,64,65]. Also, entirely asymptomatic PC [66] and non-functioning PC have been reported. Non-functioning PC usually presents at a more advanced stage with symptoms of compression or invasion in adjacent structures, such as neck mass, and/or dysphagia, hoarseness, or dyspnea [67].

In our review, seven patients (28%) were identified as asymptomatic, but three of them had a palpable neck mass on physical examination, which proved to be the PC. Also, when looking for combined clinical data (including patient family and pathological history) we identified suspicious features for PC in 64% of the patients. There were no clinical data suspicious for DTC.

### 4.3. Biological Features

There are no specific tumor markers for PC, nor for DTC, but in the case of PC malignancy it could be suspected in patients with severe hypercalcemia (>14 mg/dL or 3.5 mmol/L) and/or with marked PTH elevations (>5–10 times the upper normal limit or absolute levels >500 mg/dL) [2,65]. Non-secreting PCs with normal PTH and calcium levels have been reported in the literature [67,68,69,70]. In the cases reviewed in the current work, more than 50% had unsuspicious levels of calcium, but more than 80% of them had PTH-suggestive levels. As expected, there were no cases with suspicious elevated calcium and non-suspicious elevated PTH, leading us to believe that PTH is the most valuable laboratory finding for raising PC suspicion.

Alhough rare, non-functioning forms of PC may have normal [71] or minimally increased calcium and PTH levels [24,58,63]. Less than 10% of PC cases are non-functioning forms [24,42,72], and they are more likely to present as late-stage diseases, either due to the lack of symptoms and subtle evolution of the disease, or due to more aggressive tumor behavior [9,14]. Among the cases of associated parathyroid and thyroid carcinomas, four non-functioning PC were reported [12,21,46,52]. While the first three were either asymptomatic or without specific symptoms, the one described by D’Cruz et al. [12] had severe hypercalcemia symptoms, thus leading us to believe that only 12% (three of the 25 cases analyzed) were non-functioning. Consequently, despite the rarity of the disease, patients with slightly increased or even normal serum calcium and/or PTH should not be neglected from a suspected diagnosis of PC, especially if they present suspicious clinical features. The three non-functioning cases were not late-stage diseases, but that presented by Dikmen et al. [21] was unique: a double PC with one of the tumors localized ectopically in the thorax.

### 4.4. Imaging

Neck US, ^99m^TC-sestamibi, and high-resolution CT or MRI are valuable adjuvants for preoperative localization of pathological parathyroids, and in the case of PC and DTC for disease staging. Furthermore, careful thyroid imaging should represent a step of great importance before any surgery performed for PHPT cases [45], seeing as most synchronous parathyroid and thyroid carcinomas are diagnosed postoperatively [10].

Retrospective reviews of preoperative US indicate that PCs present features such as larger mass size than PA, heterogeneous structure, evidence of degeneration (cystic inclusions and/or calcifications), lobulated or irregular borders, and signs of local invasion [36,51,65,73,74]. Also, studies have shown that high-volume endocrine surgeons who perform systematic preoperative neck US are proficient in identifying enlarged parathyroid glands [75,76], and have the greatest potential to identify suspicious features of both PC and DTC.

Neck US is a highly sensitive method for the detection of thyroid nodules and for the evaluation of morphological features of the nodule [77,78,79], which can suggest cancer suspicion and consequently the indication of FNA [80,81] for possible cancer confirmation.

In the cases reviewed, neck US was able to identify enlarged parathyroid glands and thyroid nodules in 78% and 95% of cases, respectively. PCs were quite large, with a mean maximal diameter (during the operation) of 3.15 cm recorded, which may explain the high percentage of US-identified lesions. Parathyroids suspicious for PC were described in only half of US-identified glands, and suspicious thyroid nodules were present in three quarters of patient. Concomitant parathyroid and thyroid lesions were seen in 60% of the patients with concomitant identifiable lesions (15 patients), of whom 40% had suspicious features both for PC and thyroid carcinoma. Therefore, neck US has a good potential not only to localize the affected parathyroids and to identify concomitant thyroid nodules, but also to raise the suspicion of both PC and thyroid carcinoma, thus making it a very useful tool for determining preoperatory concomitant cancer suspicion.

When ultrasound localization fails to identify an abnormal parathyroid gland, ^99m^Tc-MIBI planar imaging or SPECT/CT may be performed to further aid preoperative localization [26], but they cannot differentiate PC from PA [82]. On the other hand, there have been false-negative results from ^99m^Tc subtraction imaging, attributed to small lesion size, abnormal ^99m^Tc uptake by the thyroid gland, and abnormal or absent ^99m^Tc uptake by a parathyroid adenoma [39,83]. Also, thyroid lesions can be sestamibi-avid, therefore ultrasound examination should always be used to increase detection of synchronous thyroid carcinomas [58,84].

In 84% of the analyzed cases, ^99m^Tc-MIBI scintigraphy was performed to localize the source of hyperparathyroidism, and 80.95% of glands were found with increased radiotracer uptake. In 17.65% of the cases, the scan was the only imagistic method able to localize the affected parathyroid gland. In 19.05%, the examinations were unable to localize the affected parathyroid gland. Of note is the case reported by Aljabri et al. [38], in which neither the scan nor neck US were able to identify the pathological parathyroid, but clinical and laboratory findings were suggestive of PC. Also to be mentioned is the complete lack in all studies of hybrid imaging evaluation (positron emission tomography with computed tomography—PET/CT or positron emission tomography with magnetic resonance imaging—PET/MRI) which might increase sensitivity, and also the very limited use of SPECT/CT (in just one case), a fact that would explain the relatively high number of negative scans.

### 4.5. FNA

Fine-needle aspiration of suspected PC should be avoided [26], not only because cytology cannot reliably differentiate between PA and PC, but more than that due to the documented risk of tumor seeding due to violation of the parathyroid capsule [23,27].

FNA of the parathyroid glands was undertaken in four cases. Three of them were thought to be parathyroid glands and one was a thyroid nodule, which on cytology showed features (marked nuclear pleomorphism with prominent nucleoli) suspicious of PC [46]. Of note, one FNA of the parathyroid gland by washout PTH was performed only to confirm the US-derived suspicion [34].

Thyroid nodule FNAs were not performed in 40% of the cases, for various reasons, some of which were obvious, including previous thyroidectomies (two cases) or no thyroid nodules identified on US (one case). On the other hand, only 10 of the 15 cases which featured US-suspicious thyroid nodules received FNA, and 70% of the cytology aspects were either benign or non-conclusive. A possible explanation is that the sensitivity of thyroid FNA can be affected by many factors including operational technique and the experience of the ultrasound specialist, the nodule size (lesser accuracy in small nodules), the sample retrieved (representative and adequate in cellularity), and the experience of the pathologist [85,86,87]. In the cases reviewed, the size of the nodules (about half of which were microcarcinomas) could have been the main source of failed FNA confirmation of cancer. Regardless of FNA results, and bearing in mind the possible concomitant association of PC and DTC, if US shows clear suspicious features of thyroid nodules on the contralateral side of the possible PC, total thyroidectomy should be performed.

In the papers reviewed, a preoperative PC suspicion was described by the authors in only 24% of cases. When we looked for possible preoperatory PC suspicion based on available clinical-–, laboratory-, US- and FNA-based data, the suspicion could have been justified in 56% of the cases.

Specific guidelines for synchronous parathyroid and thyroid carcinoma treatment are lacking. The only curative treatment for PC is en bloc resection of the affected parathyroid gland with hemithyroidectomy of the ipsilateral lobe and surrounding adherent tissue or enlarged lymph nodes identified during surgery or preoperatively diagnosed [19,20,88]. Preserving the integrity of the capsule during tumoral mass removal and complete resection of affected tissues are crucial, given that recurrent PC cases have been described due to secondary local seeding produced after rupture of the tumoral capsule and neoplastic microinfiltrations in the adjacent structures [64,89]. Thus, to properly treat PC, it is helpful for the surgeon, if possible, to form a high level of cancer suspicion before and during the operation [26]. Features such as white-greyish color, firm consistency, cystic components, and foremost the adherence to surrounding structures (especially the thyroid gland) [29,30] can aid the surgeon in raising a strong PC suspicion and performing the correct operation. Intraoperatory description of lesions revealed the highest suspicion (83.3%) of PC, although, initial en bloc resection was reported to be performed in as little as 12% of case series [26].

### 4.6. Treatment Considerations

In cases of synchronous DTC and PC there are no treatment guidelines; an appropriate strategy would be radical surgery for PC and total thyroidectomy with resection of the enlarged lymph nodes identified during surgery or preoperatively diagnosed.

En bloc resection though was performed and stated only in 52% of the cases, one of them without any suspicion of a parathyroid pathology before or during the operation. In the remaining 12 cases (48%), en bloc resection was either not performed or not mentioned, making the 52% questionable in upper and lower directions. Two of the cases without en bloc resection (Zakerkish et al. and Bednarek-Tupikowska et al.) [42,50] had previous total thyroidectomies two and six years previously, respectively. Interestingly, the postoperative outcome was in line with persistent disease in both.

The operative details of the reviewed cases also revealed a slight preponderance (60%) of left-side PCs, more than two thirds of them affecting the inferior glands.

Even though PC can be pre– and intraoperatively suspected, definitive diagnosis can only be established after complete histopathological examination [64,89,90,91]. This is based on the Shantz and Castleman criteria, used since 1973, including a trabecular pattern of the parenchymal cells, capsular and vessel invasion, high mitotic rates, nuclear atypia, and a thick capsule [82,92]. Nevertheless, histopathological diagnosis of PC can sometimes be difficult, and IHC analysis that includes parafibromin, APC, galectin-3, Cyclin D, Ki67, and other markers can aid in PC diagnosis [93,94,95,96,97]. In 28% of cases reviewed, IHC staining was additionally performed, but the reasons for performing it were not stated by the authors. Of note is the case reported by Dikmen et al. [21] with an ectopic left mediastinal parathyroid gland diagnosed and treated as concomitant PC-DTC (synchronous double PC).

Regarding the associated DTCs, histopathological examination revealed mPTC and PTC in 84% of the cases, FTC in 12% of patients, and Hurthle cell carcinoma (HCC) in one, which is in line with the common occurrence of the disease. PTC was additionally present in associations with other histological forms in 8% of the cases. Double or multiple DTCs were found in about half of patients, the majority being mPTC in their own association or with PTCs. Of note is that one PTC excision was two years before the operation for PC and concomitant mHHC, and also the case of a double FTC, the first excised five years before the removal of the PC together with the second FTC. Associated parathyroid disease was identified in 20% of the cases with equal PAs and PH, as well as one ectopic mediastinal PC.

In addition to tumoral removal of the affected gland and other involved structures, surgical treatment aims to obtain postoperative biochemical remission, with calcium and PTH normalization [38]. No consensus on follow-up for patients with PC has been established, but periodic, life-long monitoring of calcium and PTH is recommended [62,98]. Five-year survival rates vary between 85.5% [32,48] and 90.9% [53], while 10-year survival rates are reported between 49.1% [32,48] and 77% [23]. The most important prognostic factor is the successful resection of the parathyroid tumor at the time of the initial operation [58]. Postoperative normocalcemia was achieved in 72% of the patients with PC and DTC in association. Follow-up data was missing in about 25% of cases, and only three patients had a follow-up period of more than five years. The follow-up for DTC was quite uniform, complete remission being defined by clinically negative status, negative ultrasound, and negative whole body I-131 scan with nondetectable levels of thyroglobulin and anti-thyroglobulin. Considering the mentioned synchronous cancers, efficient radical treatment for DTC and facile and specific follow-up would permit the clear depiction of the diseases in cases of persistence or recurrence.

Persistent disease was present in three patients (12%), and recurrence in four (16%). All three cases with persistent hypercalcemia and three of the four cases with recurrent disease were not subjected to en bloc resection. The one case of recurrent disease with en bloc resection [45] had recurrent hypercalcemia due to a PA missed during the operation. One of the three patients with persistent disease died three weeks after surgery due to an uncontrolled hypercalcemic crisis. This was the case reported by Zakerkish et al. [42], which had some intriguing features, being the youngest age (21 years old) of all patients, with no sestamibi uptake on scintigraphy, and two years previously total thyroidectomy for HCC (the only such thyroid carcinoma variant present). Although two of the cases [39,41] had positive surgical margins on histopathological examination, only one [41] had persistent disease two years after surgery.

## 5. Conclusions

Associated PC and DTC is an exceptionally rare condition, and the preoperative diagnostic and proper treatment of patients is a substantial challenge. In most cases, pre- and intraoperative suspicious features are there to be identified. Neck US has the greatest potential to raise both DTC and PC suspicion, especially if it is performed preoperatively in a highly specialized endocrine tumor center by a multidisciplinary team. Preoperative suspicion, combined with the expert’s intraoperative identification of suspicious features, can lead to correct surgical treatment of the condition. To date, comprehensive follow-up data and centralized documentation of these rare cases have been lacking, and are of great importance to enable future studies and improvements in diagnosis and treatment.

## Figures and Tables

**Figure 1 medicina-58-01184-f001:**
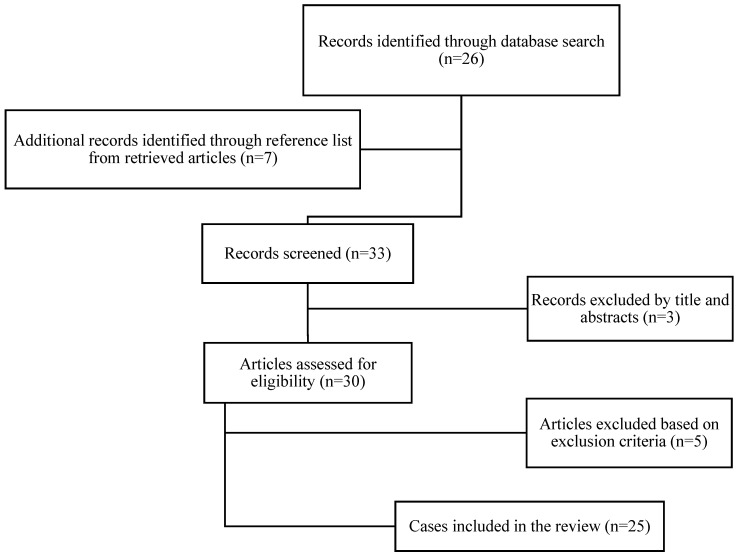
Workup for selecting relevant papers selected from the PubMed, Google Scholar, and Web of Science databases.

**Table 1 medicina-58-01184-t001:** Features of patients with synchronous parathyroid and differentiated thyroid carcinomas.

No.	Authors Year (Reference)	Age Sex	Suspicious Clinical Data EBRT+ Fh+ hPHCa+	Ca > 14 mg/dL	PTH > 5x NV	Suspicious US Parathyroid/FNA + Thyroid/FNA	^99m^TC MIBI. SPECT/CT Uptake	Possible Preop. PC Suspicion	Stated Indication for Surgery	PC Location	Intraop. PC Suspicion	En Bloc Resection PC+Thyroid	PC (cm) (g) IHC	NMTC (cm) Bilateral/Multifocal pTMN	Assoc. PA/PH	Outcome N/P/R	FLL-U (mo)
1.	**De Falco** et al., 2021 [10]	63, M	No	No	No	No + Yes (FNA − benign)	**^99m^TC MIBI.** Yes	NO	PA NG	Left inferior	No	Yes	1.2 No	2x mPTC (0.8, 0.6) pT1a(m)NxMx	No	N	84
2.	**Lam-Chung** et al., 2020 [9]	50, F	Yes **hPHCa+**	Yes	Yes	No + No	**SPECT/CT** Yes	Yes	PC suspicion –	Left superior	N/A	Yes	2.4 Yes	PTC (1.3) pT1bNxMx	No	N	1 ½
3.	**Kalthoum** et al., 2020 [33]	60, F	Yes	No	Yes	No + Yes	**^99m^TC MIBI.** Yes	Yes	PHPT –	Left superior	Yes	Yes	4 No	2x mPTC (NS) pT1a(m)NxMx	No	N	N/A
4.	**D’cruz** et al., 2020 [12]	89, F	Yes	No	No	Yes + No	**^99m^TC MIBI.** Yes	Yes	PA NG	Right inferior	Yes	Yes	1.7 Yes	FTC (3.5) pT2NxMx	PA	N	N/A
5.	**Edafe** et al., 2019 [34]	46, F	Yes **hPHCa+**	No	Yes	No + Yes (FNA − PTC)	**^99m^TC MIBI.** Yes	Yes	PC suspicion PTC	Right ?	Yes	Yes	3.3 No	PTC + mPTC (>4, NS) pT4(m)NxMx	No	N	12
6.	**Kuzu** et al., 2017 [35]	52, F	Yes **hPHCa+**	No	No	No (FNA wash-out) + Yes (FNA − benign)	**^99m^TC MIBI.** No	Yes	PA NG	Right inferior	Yes	Yes	1.8 No	PTC + mPTC (1, NS) pT1b(m)NxMx	No	N	12
7.	**Baek** et al., 2017 [36]	68, F	Yes **hPHCa+**	No	Yes	Yes + Yes (FNA − AUS/FLUS)	**^99m^TC MIBI.** Yes	Yes	PA NG	Left inferior	Yes	N/S	4.2 No	mPTC (NS) pT1a(m)NxMx	No	N	6
8.	**Demir** et al., 2017 [37]	29, F	Yes **hPHCa+**	Yes	Yes	Yes + Yes	**^99m^TC MIBI.** Yes	Yes	PA NG	Right ?	Yes	NS	2.8 No	PTC (1.6) pT1bNxMx	No	N	N/A
9.	**Aljabri** et al., 2017 [38]	72, F	Yes **hPHCa+**	No	Yes	∅ + Yes (FNA –benign)	**^99m^TC MIBI.** No	Yes	– NG	Right Inferior	Yes	Yes	4.5 Yes	mPTC (0.2) pT1a(m)NxMx	No	N	1
10.	**Dikmen** et al., 2017 [21]	57, M	No	No	No	Yes+ ∅	**^99m^TC MIBI.** Yes	Yes	Persistent elevated Ca + PTH after ePC removal	Mediastinal + left inferior	No	No	30 + 21 **(2xPC)** Yes	mPTC (0.2) pT1a(m)NxMx	ePC	N	N/A
11.	**Neslihan** et al., 2016 [39]	65, F	No	No	Yes	No + Yes (FNA − benign)	**^99m^TC MIBI.** No	Yes	PC suspicion –	Left inferior	N/A	Yes	2.5 **Positive surgical margins** Yes	2x mPTC (0.5, 0.2) pT1a(m)NxMx	No	N	N/A
12.	**Lee** et al., 2016 [40]	57, F	No	Yes	Yes	∅ + No	**^99m^TC MIBI.** Yes	Yes	PA NG	Left inferior	Yes	N/S	4.5	PTC (NS) pT?	No	-N -2xR (PC) -N	72
13.	**Song** et al., 2016 [11]	45, F	Yes	Yes	Yes	Yes + Yes	**^99m^TC MIBI.** Yes	Yes	N/S	Left inferior	Yes	No	4.3	mPTC (0.5) pT1a(m)NxMx	NA	-N -2xR (PC) -N	6mo after III surgery
14.	**Al-Sulami**, 2015 [41]	75, F	Yes	No	Yes	∅ + No	**^99m^TC MIBI.** Yes	Yes	N/S	Left ?	N/A	N/S	3.5 **Positive surgical margins**	3x mPTC (all < 0.5) pT1a(m)NxMx	N/A	P	24
15.	**Zakerkish** et al., 2015 [42]	21, M	Yes	No	Yes	No + ∅ (previous TT)	**^99m^TC MIBI.** No	Yes	PC suspected –	Right ?	N/A	No **(2 years previous TT)**	N/A	mHHC (0.6) pT1a(m)NxMx	No	P -3w later death	<1
16.	**Chaychi** et al., 2010 [43]	79, F	No **hPHCa+, 6y**	No	No	Yes + Yes (FNA − PTC)	**^99m^TC MIBI.** Yes	Yes	PHPT 2xPTC	Left superior,	N/A	Yes	5 Yes	2x PTC (2.4, 1.7) pT2(m)NxMx	No	N	6
17.	**Marcy** et al., 2009 [44]	42, F	Yes **hEBRT+**	Yes	Yes	Yes (FNA: unconclusive) + Yes (FNA − inconclusive)	**^99m^TC MIBI.** Yes	Yes	PC suspected –	Right inferior	N/A	N/S	1.3	2x mPTC (0.8, 0.5) (m)T1aNxMx	No	N	14
18.	**Goldfarb** et al., 2009 [45]	59, M	Yes **hPHCa+, 6y**	Yes	Yes	Yes + Yes	**^99m^TC MIBI.** Yes	Yes	PHPT –	Left ?	Yes	Yes	3.9 17 g	PTC + mPTC (3.2, 0.4) pT2(m)NxMx	PA	-h -R (PA) -N	14
19.	**Mazeh** et al., 2008 [46]	44, F	No	No	N/A	No + Yes (FNA − inconclusive)	N/A	**NO**	– NG	Left ?	N/A	Yes (**non-intended**)	1.5	PTC (NS) pT?	No	N	60
20.	**Lin** et al., 2005 [47]	38, M	Yes **hPHCa+, 6y**	Yes	Yes	No + Yes (FNA − PC)	**^99m^TC MIBI.** **^201^TI Scinti** Yes	Yes	PC NG	Left inferior	Yes (**frozen section!**)	No	N/A	PTC (4) pT2(m)N + Mx	No	N	72
21.	**Kern** et al., 2004 [48]	54, F	No **Fh+**	N/A	Yes	N/A	N/A	Yes	PHPT	Right inferior	Yes	No (although PC very adherent)	2.5 7 g	2x mPT + mFTC; (0.3, 0.4, 0.3) pT1a(m)N + Mx	No	N − R + 2y later (PC distant meta) − 3y death	36
22.	**Schoretsanitis** et al., 2002 [49]	55, F	Yes **hPHCa+**	Yes	Yes	Ø + No	**^+99m^TC MIBI.** Yes	Yes	PHPT NG	Left inferior	Yes	Yes	3	mPTC (0.7) pT1a(m)NxMx	PH	N	24
23.	**Bednarek-Tupikowska** et al., 2001 [50]	42, F	Yes **hPHCa+**	Yes	Yes	Yes (FNA − no cells) + Ø (previous TT)	**^99m^TC MIBI.** Yes **^99m^TC MIBI** for persistent disease No	Yes	PHPT –	Left ?	Yes	No **(6 years previous TT)**	5 Yes	FC (NS) pT?	No	P	N/A
24.	**Savil** et al., 2001 [51]	47, F	No	No	N/A	Ø + Yes (FNA − PTC)	N/A	**NO**	– PTC	Left ?	No	No	3	PTC (3) pT2(m)NxMx	PH	N	12
25.	**Kurita** et al., 1979 [52]	68, M	No	No	Yes	N/A	N/A	Yes	PHPT –	Left superior	Yes	Yes	4.2 21 g	PTC (NS) pT?	No	N	24

**Abbreviations**: hEBRT–history of external beam radiation therapy; Fh–family history; hPHCa–history of prolonged hypercalcemia, stated or with indirect signs; PT–parathyroid tumor; NMTC–non-medullary thyroid carcinoma; PHPT–primary hyperparathyroidism; PC–parathyroid carcinoma; ePC–ectopic PC; PA–parathyroid adenoma; PH–parathyroid hyperplasia; NG–nodular goiter; PTC–papillary thyroid carcinoma; mPTC–papillary thyroid micro-carcinoma; FTC–follicular thyroid carcinoma; HCC–Hurthle cell carcinoma; NV–normal value; N/S–not stated; N/A–not available; Ø–no lesion identified; IHC–immunohistochemistry; TT–total thyroidectomy; FLL-U (mo)– follow-up (months); Outcome N/P/R–normocalcemic/persistence/recurrence.

**Table 2 medicina-58-01184-t002:** Parathyroid diseases associated to PC.

Parathyroid Disease	PA	PH	Ectopic PC
**No./total (%)**	2/5 (40)	2/5 (40)	1/5 (20)
**Calcium levels mg/dL**	13.8; 14.7	11.3; 12.1	16.2
**PTH levels pg/mL**	441; 318	197; 249	4211

**Abbreviations**: Parathyroid adenomas (PA) and parathyroid hyperplasia (PH). Ectopic parathyroid carcinoma (PC).

## Data Availability

The authors confirm that the data supporting the findings of this study are available within the article.

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
