# Peer review of "Association of Parathyroid and Differentiated Thyroid Carcinomas: A Narrative Up-To-Date Review of the Literature"

_medicina, 2022, doi:10.3390/medicina58091184_

Round 1

Reviewer 1 Report

This work has "complementary" elements in its particular area of interest. 

Congratulations to the authors for their efforts.

Comments:

*The information about Parathyroid and Thyroid seems to be intertwined, making the reading process a little difficult. The information is correct and enough. My minor recommendation; it would be more reader-friendly to re-organize the section in the Background/Introduction heading.

*Abbreviations in the abstract are repeated in the introduction section. page 2 paragraph 2; PC, DTC, PTC

*page 3, paragraph 2->"undergone_en-bloc resection" typing error

*Page 3, paragraph 4; correct the last part of this sentence "Due to the rarity of ........, and no randomized extensive studies"

*page 3, paragraph 5, that starts with "The aim of the study" replace files->field, picture of challenges->picture of the challenges.

*Please clarify the "the earliest record" date in the methods section. If different between databases provide them separately.

*Exclusion and Inclusion criteria are well defined. 

*In the first diagram in Figure 1, you can add databases in parentheses to the "database search" section. Or in the figure description. Sometimes a single schema can summarize the entire method.

*A space should be left between the age and sex column title in Table 1.

*Suspicious elevated serum calcium levels (14 mg/dl) was  were found....

*such as alkaline phosphatse, phosphatase.......

*page 11, paragraph 5
There was a preponderance for of left side parathyroid lesions (12 cases, 60%), of which more than 2/3 were inferior glands. Of note, is one case with an ectopic left mediastinal parathyroid gland that was diagnosed and operated on in the recent past of the concomitant PC-NMTC operation (synchronous double PC). 

*In the discussion part, references should be corrected
example: "published review by de Falco et al. [Error! Bookmark not defined.]..."

*The discussion section contains intensive information. I would like to suggest that it will be more convenient for the reader to compile some of the parts in small titles.

Such as clinical manifestations of PC and DTC or Neck US and FNA,...

Author Response

Respecter Reviewer 1,

Thank you very much for your thorough review of the manuscript. Please find below the resolution to your suggestions.

*The information about Parathyroid and Thyroid seems to be intertwined, making the reading process a little difficult. The information is correct and enough. My minor recommendation; it would be more reader-friendly to re-organize the section in the Background/Introduction heading.

Response: Background/Introduction section has been re-organized in order to ease the reading.

*Abbreviations in the abstract are repeated in the introduction section. page 2 paragraph 2; PC, DTC, PTC

Response: PC, DTC, PTC, US and FNA abbreviations in page 2, paragraph 2 have been corrected and are now not repeating from abstract.

*page 3, paragraph 2->"undergone_en-bloc resection" typing error

Response: Typing error corrected into “en bloc resection”.

*Page 3, paragraph 4; correct the last part of this sentence "Due to the rarity of ........, and no randomized extensive studies"

Response: we have made the corrections accordingly.

*page 3, paragraph 5, that starts with "The aim of the study" replace files->field, picture of challenges->picture of the challenges.

Response: we have made the corrections accordingly.

*Please clarify the "the earliest record" date in the methods section. If different between databases provide them separately.

Response: First record was identified in 1979 (Kurita et al., 1979) and the year was added in the text.

*Exclusion and Inclusion criteria are well defined. 

Response: Thank you for the appreciation.

*In the first diagram in Figure 1, you can add databases in parentheses to the "database search" section. Or in the figure description. Sometimes a single schema can summarize the entire method.

Response: We have included in the Figure description the databases used for the articles selections. Description is now as follows: “Workup for selecting relevant papers selected from the PubMed, Google Scholar and Web of Science databases”.

*A space should be left between the age and sex column title in Table 1.

Response: we have corrected accordingly.

*Suspicious elevated serum calcium levels (14 mg/dl) was  were found....

Response: we have corrected accordingly.

*such as alkaline phosphatse, phosphatase.......

Response: we have corrected accordingly.

*page 11, paragraph 5
There was a preponderance for of left side parathyroid lesions (12 cases, 60%), of which more than 2/3 were inferior glands. Of note, is one case with an ectopic left mediastinal parathyroid gland that was diagnosed and operated on in the recent past of the concomitant PC-NMTC operation (synchronous double PC). 

Response: we have made the corrections accordingly.

*In the discussion part, references should be corrected
example: "published review by de Falco et al. [Error! Bookmark not defined.]..."

Response: De Falco et al. corresponds to reference 10, and correction has been made.

*The discussion section contains intensive information. I would like to suggest that it will be more convenient for the reader to compile some of the parts in small titles.

Such as clinical manifestations of PC and DTC or Neck US and FNA,...

Response: The following subtitles were addes in the Discussion section in order to facilitate the reading:

  • Etiopathology of PC and DTC;
  • Clinical manifestations;
  • Biological features;
  • Imaging;
  • FNA;
  • Treatment considerations

Reviewer 2 Report

The authors have done nice job to summarize the literature review to identify the association between parathyroid and thyroid carcinomas. However, following comments will make this review article better and comprehensive.

1. Search latest literature and may help you to add more cases.

2. The references cited and reference style needs to be modified to make it even and according to the journal recommendations.

3. Do final check of English and scientific language.

Author Response

Respecter Reviewer 2,

Thank you very much for your review of the manuscript. Please find below the resolution to your suggestions.

  1. Search latest literature and may help you to add more cases.

Response: we have searched the literature up to August 2022. No new entries were found, therefor our database remains the same, still we have modified in text in the section Methods, 1st paragraph as follows: “We performed a comprehensive search of several databases such as PubMed, Google Scholar and Web of Science from the earliest record in 1979 to August 2022.”

  1. The references cited and reference style needs to be modified to make it even and according to the journal recommendations.

Response: In text reference and reference list was modified according to the journal recommendations.

  1. Do final check of English and scientific language.

Response: A final English language writing correction was made.